# 5α-cyprinol sulfate, a bile salt from fish, induces diel vertical migration in *Daphnia*

**Meike Anika Hahn[1]\*, Christoph Effertz[1], Laurent Bigler[2], Eric von Elert[1]**

[1]Aquatic Chemical Ecology, Department of Biology, University of Koeln, Koeln, Germany; [2]Department of Chemistry, University of Zurich, Zurich, Switzerland

**Abstract** Prey are under selection to minimize predation losses. In aquatic environments, many prey use chemical cues released by predators, which initiate predator avoidance. A prominent example of behavioral predator-avoidance constitutes diel vertical migration (DVM) in the freshwater microcrustacean *Daphnia* spp., which is induced by chemical cues (kairomones) released by planktivorous fish. In a bioassay-guided approach using liquid chromatography and mass spectrometry, we identified the kairomone from fish incubation water as 5α-cyprinol sulfate inducing DVM in *Daphnia* at picomolar concentrations. The role of 5α-cyprinol sulfate in lipid digestion in fish explains why from an evolutionary perspective fish has not stopped releasing 5α-cyprinol sulfate despite the disadvantages for the releaser. The identification of the DVM-inducing kairomone enables investigating its spatial and temporal distribution and the underlying molecular mechanism of its perception. Furthermore, it allows to test if fish-mediated inducible defenses in other aquatic invertebrates are triggered by the same compound.
DOI: https://doi.org/10.7554/eLife.44791.001

## Introduction

Predation is recognized as an important selective force which has been shown to drive the shape of trophic cascades and numerous aspects of ecosystem ecology (*Gliwicz, 1986*; *Lima, 1998*; *Werner and Peacor, 2003*). Such effects of predators are caused by predation (direct consumptive effect; *Preisser et al., 2005*) or by effects on plastic behavioral, physiological or morphological traits of prey seeking to avoid predation (indirect, non-consumptive effect, (*Heithaus et al., 2008*; *Lima, 1998*; *Moll et al., 2017*; *Werner and Peacor, 2003*). The concept has been introduced that prey species must effectively balance consuming their resources against becoming resources for their predators (*Abrams, 1984*; *Sih, 1980*) in order to maximize their own fitness. It has become clear that non-consumptive predator effects strongly affect prey distribution, demography and behavior (*Heithaus et al., 2008*; *Lima, 1998*; *Werner and Peacor, 2003*) and the strength of top-down and bottom-up control in communities (*Ford et al., 2014*). Hence, non-consumptive predator effects are mediated by shifts in plastic traits of prey, and in aquatic food webs such non-consumptive effects exceed direct consumptive effects (*Preisser et al., 2005*; *Turner et al., 2000*).

Such adaptive shifts in prey traits require an accurate assessment of the predation risk, which may be accomplished by physical contact, vision or by chemical cues. Chemical cues are superior cues in turbid or dark environments (*Kats and Dill, 1998*) or in cases in which the escape capability upon attack is low in prey, for example due to low escape velocity. In line with this, the induction of defenses by chemical cues from predators is widespread in aquatic systems (*Bjærke et al., 2014*; *Brönmark and Hansson, 2012*), and recently progress has been made by identification of chemical cues involved in aquatic predator-prey chemical communication (*Poulin et al., 2018*; *Selander et al., 2015*; *Weiss et al., 2018*).

One classical example of behavioral predator avoidance is diel vertical migration (DVM) in the freshwater microcrustacean *Daphnia*, which play a key role in lakes and ponds, as they are major

**\*For correspondence:**
meike.hahn@uni-koeln.de

**Competing interests:** The authors declare that no competing interests exist.

consumers of planktonic primary producers and important prey for higher trophic levels (*Miner et al., 2012*). DVM is a widespread predator avoidance behavior (*Hays, 2003*; *Williamson et al., 2011*), in which the exposure to UV (*Rhode et al., 2001*) and the risk of predation by visually oriented predators, such as fish, is reduced by daytime residence in the dark, deep layer of the water column (*Hays, 2003*; *Stich and Lampert, 1981*). At night, zooplankton emerges from the depth into the upper water layer to minimize demographic costs associated with residence in the cold, food-depleted deep-water refuge (*Loose and Dawidowicz, 1994*; *Stich and Lampert, 1981*). Hence, DVM of *Daphnia* negatively affects the foraging success of planktivorous fish. This behavioral anti-predator defense affects the control of planktonic primary producers by zooplankton in the open water and thus impacts many other ecosystem-wide processes (*Haupt et al., 2010*; *Reichwaldt and Stibor, 2005*).

In lakes and ponds, DVM in zooplankton is triggered by changes in light intensity (*Effertz and von Elert, 2014*; *Ringelberg, 1999*) and by chemical cues released by predators (*Dodson, 1988*; *Lampert, 1993*; *Neill, 1990*). Such chemical cues are termed kairomones, if they mediate a transfer of information among species that imparts a benefit to the receiving organism while not being beneficial for the producer (*Pohnert et al., 2007*). The finding that the amplitude of DVM increases with fish density indicates that kairomones provide a reliable indicator for the risk of *Daphnia* of being preyed upon by fish (*Van Gool and Ringelberg, 2002*; *von Elert and Pohnert, 2000*). Thus, fish - *Daphnia* interactions are to a large degree determined by indirect, non-consumptive effects of fish that are mediated by kairomones inducing predator avoidance by DVM.

Lakes and ponds are the best available freshwater source providing ecosystem services like domestic, industrial and recreational usage of water bodies (*Rudman et al., 2017*). Global warming and ongoing nutrient input are predicted to deteriorate water quality and negatively affect ecosystem services of lakes and ponds (*Huisman et al., 2018*; *Rudman et al., 2017*), which increases the need for succesful lake management. One tool frequently used in lake management is food chain manipulation, which aims at manipulating the interaction of *Daphnia* and fish (*Jeppesen et al., 2017*; *Peretyatko et al., 2012*; *Søndergaard et al., 2017*). However, at the molecular level this interaction is not fully understood unless the chemical nature of the fish kairomone is disclosed. Earlier we have shown that the release of the DVM-inducing kairomone does not depend on prior feeding of fish (*Loose et al., 1993*). The kairomone can be extracted from fish incubation water by lipophilic solid phase extraction, and we have characterized it as a low molecular weight compound carrying a hydroxyl group and a negative charge (*Loose et al., 1993*; *von Elert and Pohnert, 2000*; *von Elert and Loose, 1996*). However, the exact chemical identity of the fish kairomone remained unknown. Here, we use the microcrustean *Daphnia*, which has become a model organism for the induction of DVM in response to fish (*Lampert, 2006*; *Miner et al., 2012*), to identify the yet unknown kairomone.

## Results

In order to ensure identification of the major infochemical, we used a bioassay-guided approach. DVM was assessed using a well-established indoor setup in which daytime residence depth in a temperature stratified water column is determined as a proxy for DVM of *Daphnia* (*Brzeziński and von Elert, 2015*; *Loose et al., 1993*; *von Elert and Pohnert, 2000*). It has earlier been shown that the DVM-inducing activity of fish incubation water can be efficiently extracted by lipophilic solid-phase extraction, whereas solid-phase extracts of control water were not active (*von Elert and Loose, 1996*). We followed the activity in such an extract of fish incubation water (EFI) of roach (*Rutilus rutilus*, Cyprinidae) through separation by HPLC and by investigating fractions that cover the whole chromatogram. EFI was found to be active, as it induced a significant shift in the daytime residence depth of *Daphnia*. Fractionation of EFI by HPLC yielded six fractions (*Figure 1A*), of which only fraction three proved to induce DVM (*Figure 1B*, *Figure 1—source data 1*).

Since the kairomone has been shown to be an anion (*von Elert and Loose, 1996*), LC high-resolution electrospray mass spectrometry (HR-ESI-MS) was performed in the negative ionization mode. Mass data extracted from the time window corresponding to the active fraction 3 (11–14 min), indicated that in this active fraction the most abundant ion had a *m/z* of 531.29993 corresponding to the deprotonated molecule [M– H]⁻, which led to $M_{calc.}$ = *m/z* 532.3070 (*Figure 2A*). The collision-induced fragmentation of [M– H]⁻ resulted in the detection of the precursor ion and a product ion

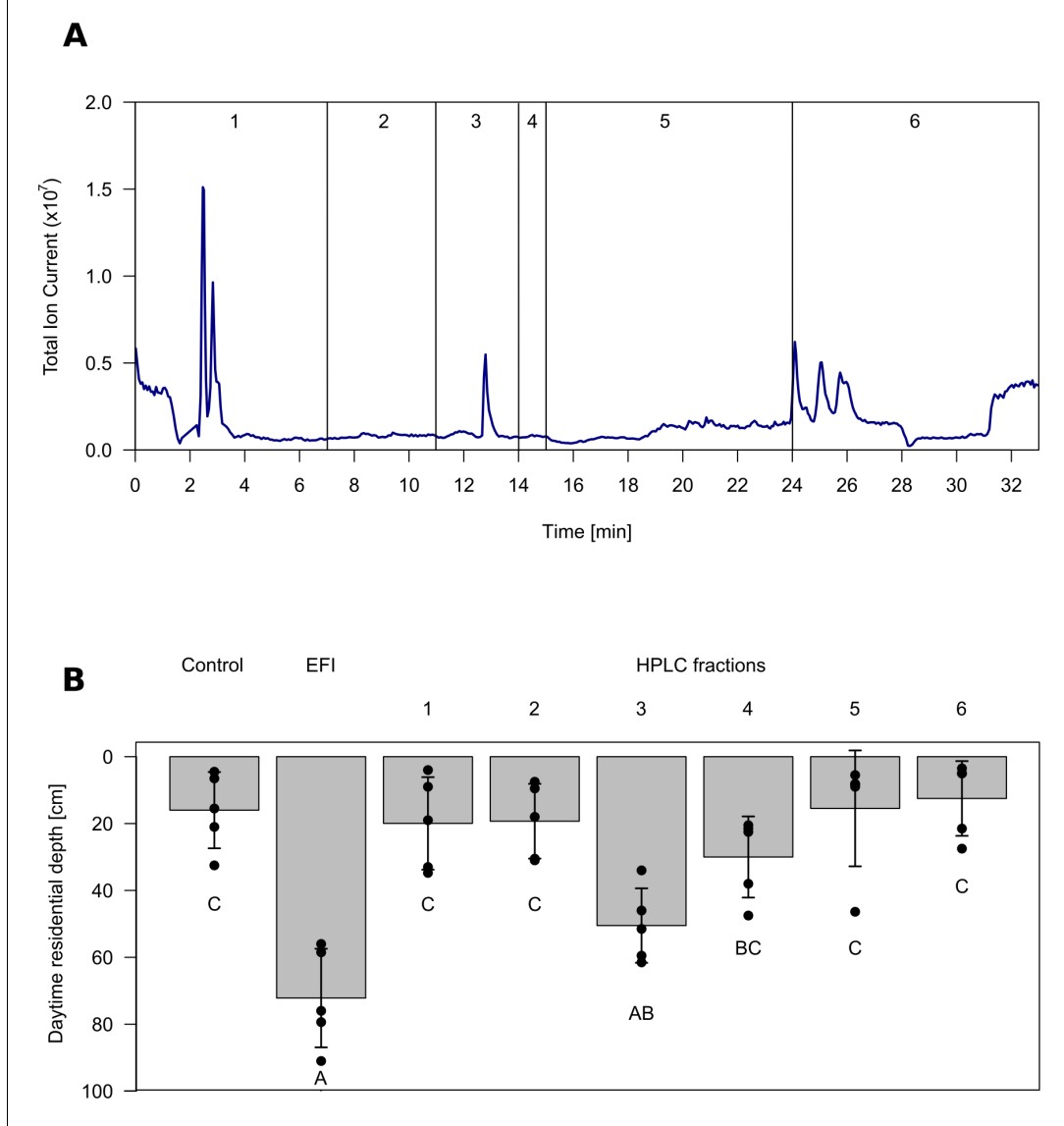

**Figure 1.** Chromatogram and biological activity of extracted fish incubation water (EFI). (**A**) Chromatogram of EFI after separation on a 250 mm x 4 mm reversed phase column (Nucleosil 100–5 $C_{18}$, Macherey-Nagel, Düren, Germany) using the ammonium acetate buffered mobile phases A $H_2O$ and B acetonitrile/MeOH (13:6, (v/v) with the portion of B increasing over time. Vertical lines and numbering indicate collection of fractions 1–6. (**B**) Behavioral response of *Daphnia magna* to extract of fish incubation water (EFI) and fractions thereof. The control contains the same volume of pure organic solvent as tested from EFI and its fractions. Mean *Daphnia* daytime residence depth (± SD, n = 4). Different capital letters indicate significant differences among treatments after one-way ANOVA followed by Tukey's HSD test. Statistical results are summarized in *Figure 1—source data 1*.
DOI: https://doi.org/10.7554/eLife.44791.002

The following source data and figure supplement are available for figure 1:

**Source data 1.** Temperature profile of the experimental tubes.
DOI: https://doi.org/10.7554/eLife.44791.004
**Source data 2.** Response of *Daphnia* to HPLC fractions.
DOI: https://doi.org/10.7554/eLife.44791.005
**Figure supplement 1.** Temperature profile of the experimental tubes.
DOI: https://doi.org/10.7554/eLife.44791.003

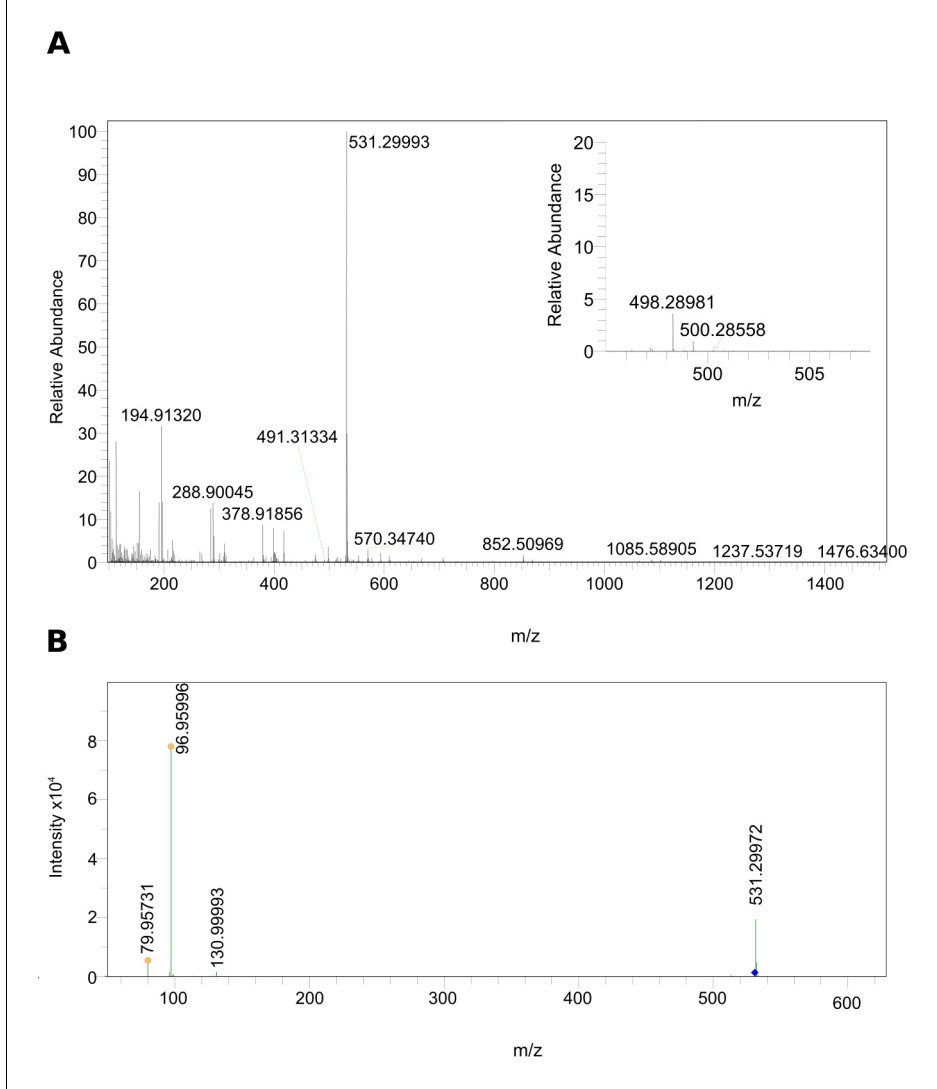

**Figure 2.** MS data of the biologically active HPLC fraction. (**A**) ESI-MS spectrum extracted from the time window corresponding to the active fraction three in *Figure 1* (11 - 14 min) measured in the negative ionization mode. m/z 531.29986 [M–H]⁻ indicates the presence of 5α-cyprinol sulfate (α-CPS), and m/z 498.28981 [M–H]⁻ (see insert) indicates the presence of the conjugated bile acid taurochenodeoxycholic acid (TCDCA). (**B**) Negative ionization MS-MS of m/z 531.29986 [M–H]⁻: 96.95996 ($HSO_4^-{}_{calc}$=m/z 96.96010, Δ = –1.44 ppm), which indicated the presence of a sulfate group in m/z 531.29986 [M–H]⁻.

DOI: https://doi.org/10.7554/eLife.44791.006

with a mass of m/z 96.95996 (*Figure 2B*) corresponding to the monoisotopic mass of hydrogen sulfate ($HSO_4^-{}_{calc}$=m/z 96.96010, Δ = –1.44 ppm), thus indicating the presence of a sulfate group. The tool sCLIPS included in the software MassWorks (cerno BIOSCIENCE) was used to predict the corresponding sum formula of the detected molecule based on the detected peak of its monoisotopic mass as well as its isotope pattern. The predicted sum formula for [M-H]⁻ was $C_{27}H_{48}O_8S$. A data base search using the platform PubChem yielded as best hit cyprinol sulfate (CPS), a known bile compound of fish (*Hagey et al., 2010*), as a candidate compound for the DVM-inducing activity. This finding was well in accordance with the detected sulfate group.

Subsequent search for other known bile compounds of fish in the active fraction three revealed the presence of taurochenodeoxycholic acid (TCDCA, *Figure 2A*, insert, [M-H]⁻=m/z 498.28981, $C_{24}H_{44}NO_6S$, $M_{calc}$ = m/z 499.28981, Δ=1.8 ppm).

As CPS and TCDCA are known constituents of fish bile (*Hagey et al., 2010*), we tested bile, which was obtained from the gallbladder of common carp (*Cyprinus carpio*, Cyprinidae) and rainbow trout (*Oncorhynchus mykiss*, Salmonidae) for DVM induction in *Daphnia*. Dilutions of extracted bile of both species corresponding to similar CPS concentrations as found in EFI (carp: 1 and 2 nM, trout: 0.4 and 3.9 nM, EFI: 1 nM) proved to induce DVM with amplitudes that were statistically not distinguishable from those induced by EFI (*Figure 3—figure supplement 2* and *Figure 3—figure supplement 2—source data 1*).

When we compared chromatograms of EFI and the fish bile samples, identical retention times, high-resolution *m/z ratios* and fragmentation patterns after MS-MS confirmed that α-CPS and TCDCA were also present in carp bile and trout bile (*Figure 3*, *Figure 3—figure supplements 1* and *2*). It is well known, that 5α-cyprinol sulfate (α-CPS) constitutes the major bile compound in cyprinid fish (*Hagey et al., 2010*). Since it is not commercially available, we have isolated α-CPS on a semi-preparative scale from gallbladder of common carp (*Cyprinidae*) and determined its structure by NMR to be α-CPS in a previous study (*Hahn et al., 2018*). Hence we conclude that *m/z* 531.29986 indicates the presence of α-CPS in EFI. The chemical identity of α-CPS and TCDCA was further confirmed by chromatography of reference compounds, which showed retention times and high-resolution *m/z* values that matched those in the chromatogram of EFI (*Figure 3*). We conclude that the major constituent of the only active fraction in EFI was the bile salt α-CPS. One further abundant bile salt in this fraction was TCDCA. While the ion detected at 2,04 min is of unknown identity (*Figure 3C*), we speculate that the second peak in the EIC of *m/z* = 531.29986 at 2.38 min (*Figure 3D*) is the isomer β-cyprinol sulfate.

α-CPS isolated from bile of common carp was used for dose response experiments for the induction of DVM (*Figure 4*) after checking its purity (*Figure 4—figure supplements 1* and *2*). α-CPS induced the same predator-avoidance behavior as extracted fish incubation water (EFI) at concentrations ≥ 100 pM (Figur 4A, *Figure 4—source data 2*). TCDCA proved to be active at ≥20 μM only (*Figure 4B*, *Figure 4—source data 3*). As a possible microbial degradation product of TCDCA , we further tested chenodeoxycholic acid (*Shimada et al., 1969*) which proved to be inactive with respect to DVM-induction in concentrations ≤ 25 μM (*Figure 4C*, *Figure 4—source data 4*). In other words, *Daphnia* were five orders of magnitude more sensitive to α-CPS than to TCDCA, while no DVM induction was found in response to chenodeoxycholic acid (CDCA), the deconjugated form of TCDCA.

We quantified α-CPS and TCDCA in extracted fish incubation water (EFI, *Figure 4—figure supplement 3*). As 20 μl of EFI are derived from 1 L of fish incubation water, this volume was dissolved per liter to obtain biologically active water in the bioassay. The actual concentration of α-CPS in biologically active water was 1.03 ± 0.08 nM (mean ± SD, n = 3) and that of TCDCA was 0.14 ± 0.01 nM (mean ± SD, n = 3). Hence the concentration of α-CPS was 7.5-fold higher than that of TCDCA. This concentration of α-CPS in biologically active water clearly exceeded the threshold concentration for biological activity of 100 pM, whereas TCDCA was present in concentrations that were more than five orders of magnitude below the activity threshold concentration of 20 μM. We therefore conclude that α-CPS is the only tested bile salt that is relevant for the induction of DVM in *Daphnia* in response to fish.

## Discussion

We identified an infochemical that *Daphnia* uses to perceive the risk of being preyed upon by planktivorous fish. We show that the induction of DVM in *Daphnia* is dependent on a particular bile compound from fish, that is 5α-cyprinol sulfate (α-CPS), which is active already at a concentration of 100 pM. The identification of α-CPS is the result of a bioassay-guided approach in which we have targeted the main activity throughout all purification steps. Hence, α-CPS constitutes the major activity in fish incubation water. Although another bile compound (TCDCA) was found in the active fraction obtained by HPLC, only α-CPS was present in sufficiently high concentrations to be biologically active. This is due to two factors: (i) *Daphnia* is five orders of magnitude more sensitive to α-CPS than to TCDCA and (ii) the concentration of α-CPS in extracted fish incubation water is approximately sevenfold higher than that of TCDCA. This substantially higher concentration of α-CPS is well in accordance with the findings that α-CPS constitutes the major bile salt in cyprinid bile (*Hofmann et al., 2010*).

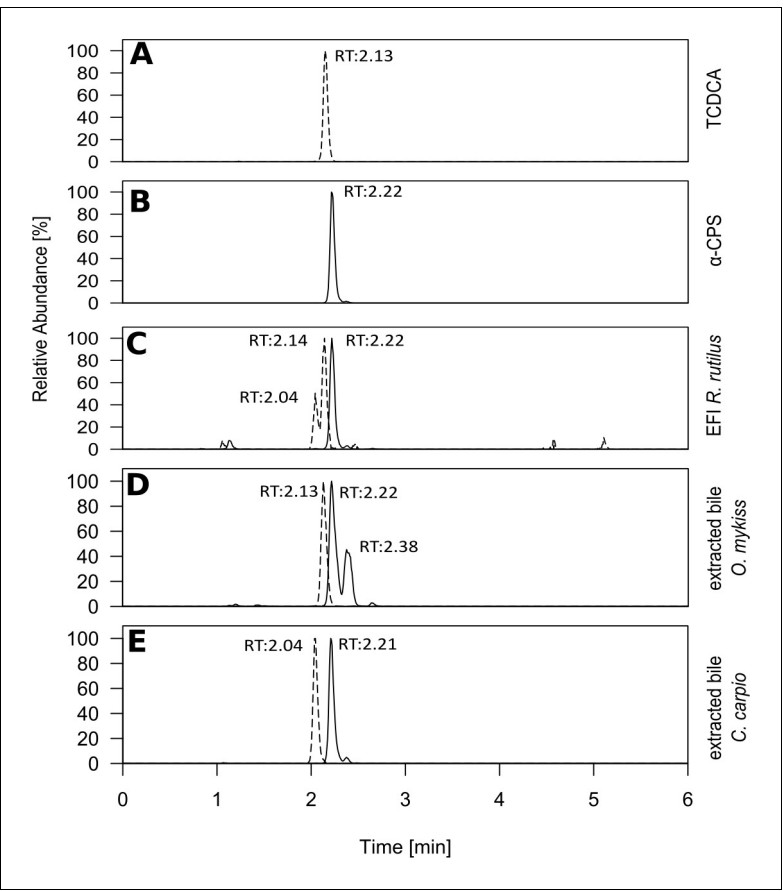

**Figure 3.** Retention times of selected bile salts in extracted fish incubation water and fish bile. Extracted ion chromatograms of cyprinol sulfate ([M]$^-_{calc}$=$m/z$ 532.3070; black line) and the conjugated bile acid taurochenodeoxycholic acid (TCDCA) ([M]$^-_{calc}$=$m/z$ 498.2889; dashed line) after ESI-MS in the negative ionization mode, which were present in the active fraction of extracted fish incubation water (*Figure 2A&B*). (A) pure TCDCA and (B) reference substances for 5α-cyprinol sulfate from *Hahn et al. (2018)* (5α-CPS). Based on identical retention time with B, α-CPS purified from carp bile and due to MS/MS (*Figure 3—figure supplement 1*) 5α-CPS is present in extracted fish incubation water (EFI) from *Rutilus rutilus* (C) and in extracted bile from *Oncorhynchus mykiss* (D) and from *Cyprinus carpio* (E). Relative intensities, with the highest set to 100%, are depicted.

DOI: https://doi.org/10.7554/eLife.44791.007

The following source data and figure supplements are available for figure 3:

**Figure supplement 1.** MS-MS (negative ionization mode) of 531.2997 [M-H]$^-$: 96.95980–96.95992 at 50 eV for (A) 5α-CPS purified from carp bile, (B) extracted fish incubation water (EFI) of *R.rutilus* and (C) extracted bile of rainbow trout *Oncorhynchus mykiss* for the time windows, when α-CPS (M$_{calc}$ = $m/z$ 531.29986) was detected during LC-MS of *Figure 3*.

DOI: https://doi.org/10.7554/eLife.44791.008

**Figure supplement 1—source data 1.** MS-MS measurements of m/z=531.2997 deriving from fish incubation water and fish bile.

DOI: https://doi.org/10.7554/eLife.44791.009

**Figure supplement 2.** Behavioral response of *Daphnia magna* to extract of fish incubation water (EFI) and to extract of carp and trout bile.

DOI: https://doi.org/10.7554/eLife.44791.010

**Figure supplement 2—source data 1.** Response of *Daphnia* to extracts of fish incubation water and fish bile.

DOI: https://doi.org/10.7554/eLife.44791.011

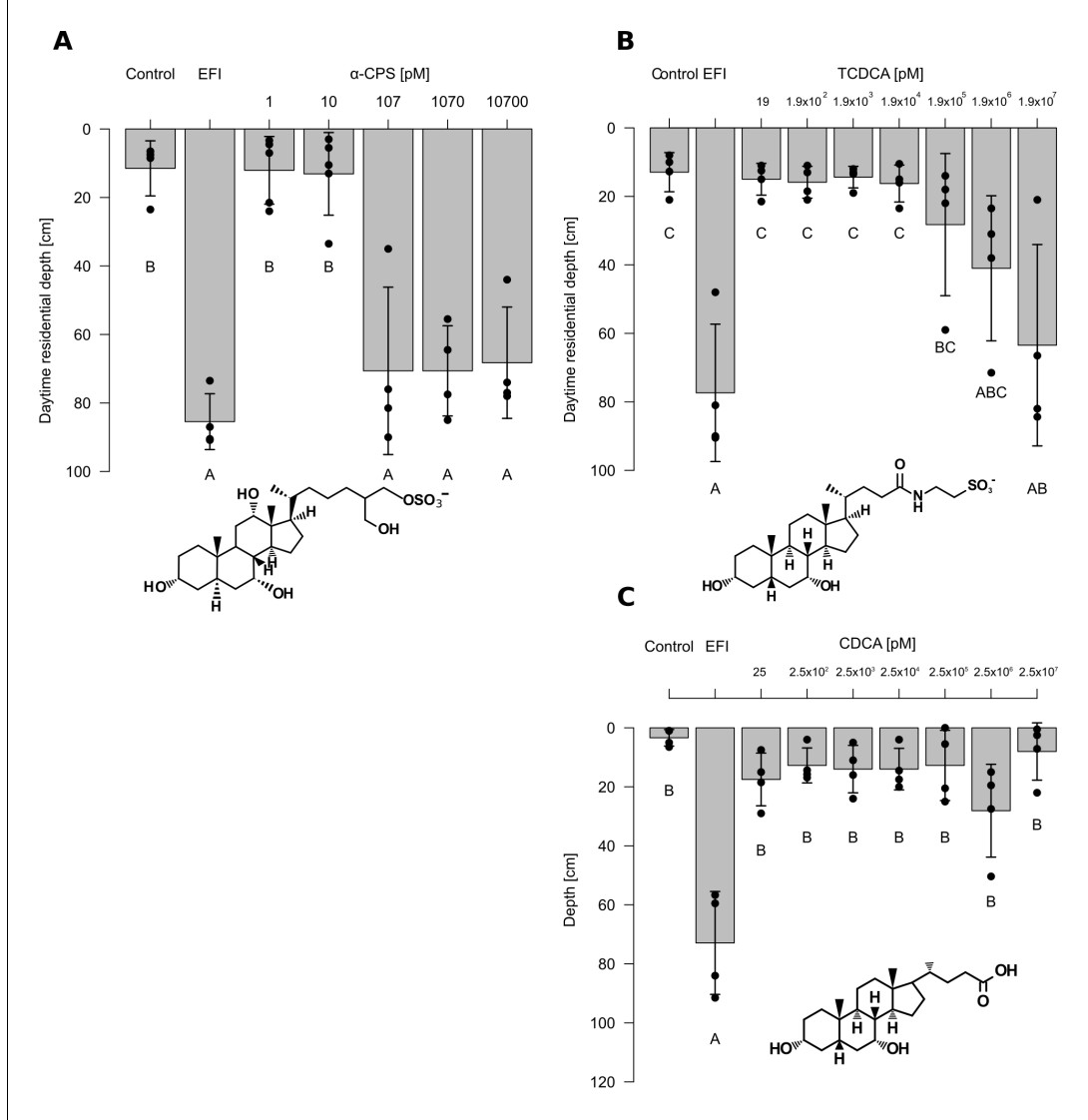

**Figure 4.** Effects of extracted fish incubation water (EFI) and of increasing concentrations of selected bile salts. (A) 5α-cyprinol sulfate (5α-CPS), (B) taurochenodeoxycholic acid (TCDCA), and. (C) chenodeoxycholic acid (CDCA) on behavioral responses of *Daphnia*. Controls contain equivalent volumes of pure organic solvent. Mean *Daphnia* daytime residence depth (± SD, n = 3–4) is depicted. The day:night cycle was 16:8 hr. Different capital letters indicate significant differences among treatments (one-way ANOVA followed by Tuke's HSD test). Statistical results are summarized in *Figure 4—source data 2–4*.

DOI: https://doi.org/10.7554/eLife.44791.012

The following source data and figure supplements are available for figure 4:

**Source data 1.** Effects of extracted fish incubation water (EFI) and of increasing concentrations of selected bile salts on diel vertical migration.

DOI: https://doi.org/10.7554/eLife.44791.017

**Source data 2.** Response of *Daphnia* to 5α-cyprinol sulfate.

DOI: https://doi.org/10.7554/eLife.44791.018

**Source data 3.** Response of *Daphnia* to 5α-cyprinol sulfate.

DOI: https://doi.org/10.7554/eLife.44791.019

**Source data 4.** Response of *Daphnia* to chenodeoxycholic acid (CDCA).

DOI: https://doi.org/10.7554/eLife.44791.020

**Figure supplement 1.** Total ion chromatogram of 5α-cyprinol sulfate (α-CPS) purified from fish bile (*Cyprinus carpio*).

DOI: https://doi.org/10.7554/eLife.44791.013

**Figure supplement 2.** MS data of the subfraction from 1 to 3 min of LC-MS runs of purified 5α-cyprinol sulfate (α-CPS) and a blank sample (SI *Figure 4*).

DOI: https://doi.org/10.7554/eLife.44791.014

eLIFE Research article

Ecology

*Figure 4 continued*

**Figure supplement 3.** Calibration curves after ESI-MS of (**A**) taurochenodeoxycholate (TCDCA) and (**B**) cholesteryl sulfate used for the quantification of 5α-cyprinol sulfate.

DOI: https://doi.org/10.7554/eLife.44791.015

**Figure supplement 3—source data 1.** Calibration curves of taurochenodeoxycholate and cholesteryl sulfate.

DOI: https://doi.org/10.7554/eLife.44791.016

Concerning the discussion of the estimated threshold concentration of α-CPS for DVM induction one methodological detail has to be considered. α-CPS present in EFI was quantified as equivalents of cholesteryl sulfate, since there has not been enough material of α-CPS to conduct an absolute gravimetric quantification. The quantification of α-CPS by means of a calibration curve of cholesteryl sulfate can however be considered quite accurate since both molecules propably result in comparable detector responses of the ESI-MS for the following reasons: i) both α-CPS and cholesteryl sulfate carry a sulfate group and, without prior ionization, are already negatively charged when reaching the ESI source. Different ionizabilities of both analytes can thus be excluded as source of inaccuracy. ii) The $m/z$ ratios of α-CPS and cholesteryl sulfate are so close to each other ($m/z$ = 465.3039 vs. $m/z$ = 531.2992), that tuning of the mass spectrometer's ion optics will not differently affect the detectability of either compound. iii) Just like the $m/z$ ratios, the molecular masses of the molecules are comparable as well, so that differences with respect to spray formation in the ESI source are probably of minor importance. In conclusion, it is reasonable to assume that quantification of α-CPS as equivalents of cholesteryl sulfate results in minor inaccuracies only.

α-CPS matches perfectly the previously published chemical characteristics for the DVM-inducing fish kairomone of roach (*Rutilus rutilus*) and other cyprinid fishes (*Loose et al., 1993*; *von Elert and Loose, 1996*). Its molecular mass is close to 500 Dalton, it contains hydroxyl and no amino groups, has a strong negative charge and, as predicted, α-CPS is free of olefinic bonds. The anionic character and the hydroxyl groups make α-CPS a good infochemical in water, as they render the molecule water soluble so that it can well diffuse after release by the predator.

α-CPS, a sulfated bile alcohol, belongs to the bile salts, which are metabolites of cholesterol in vertebrates with the primary function to emulsify dietary fats and to facilitate their intestinal absorption (*Maldonado-Valderrama et al., 2011*). Although bile salts are an essential component of the enterohepatic circulation, fish excrete bile salts via the intestine, the gills and the urinary tract (*Buchinger et al., 2014*). Excreted fish bile salts are often highly water soluble due to conjugation with a sulfate (in case of bile alcohols), or with taurine (in case of bile acids). Furthermore, bile salts are generally stable as they must be resistant to digestion within an organism, which makes conjugated bile salts ideal infochemicals. Corroborating results for other bile salts (*Zhang et al., 2001*), we here demonstrate the release of α-CPS by starved fish. Hence, the induction of DVM by fish does not require successful hunting of fish. The fact that bile salts are essential in fish metabolism explains why, from an evolutionary perspective, fish have not stopped to release bile salts despite its disadvantage for the emitter organism.

The DVM-inducing kairomone that we report here is to our knowledge the first identification of an aquatic kairomone that mediates predator-prey interactions between zooplankton and fish. The other few cases in zooplankton, for which chemical cues have been identified, are interactions of herbivorous zooplankton with its phytoplankton prey (*Selander et al., 2015*; *Uchida et al., 2008*) and with an invertebrate predator (*Weiss et al., 2018*): Copepods were shown to release copepodamides, a class of taurine containing polar lipids, which leads to an increased toxicity in their dinoflagellate prey (*Selander et al., 2015*); *Daphnia* have been reported to release an alkylsulfate into the water that induces the formation of protective colonies in the green alga *Acutodesmus obliquus* (*Uchida et al., 2008*), and larvae of *Chaoborus* sp. release fatty acids bound to a particular amino acid, which lead to changes in morphology in *Daphnia pulex* (*Weiss et al., 2018*). Among benthic animals a predatory crab has been shown to release two pyridin-metabolites that induce defensive behavior in a prey crab (*Poulin et al., 2018*). The sterane core of α-CPS renders it partly lipophilic, and causes together with the sulfate group the overall amphipathic nature of α-CPS. Sulfate or sulfonium groups are present in several of the compounds known in chemical communication of zooplankton (*Selander et al., 2015*; *Uchida et al., 2008*) and as well in the sex pheromone of lamprey (*Li et al., 2002*).

We accomplished the identification of α-CPS as the DVM-inducing kairomone using incubation water from *Rutilus rutilus* (Cyprinidae). The family of Cyprinidae is with more than 2000 species the largest family of fishes, and dominates freshwater habitats around the world with the exception of South America and Australia, and a considerable number of species is found in brackish water (*Hastings et al., 2015*). In line with the high diversity of cyprinid fish, incubation water of many cyprinid species has been demonstrated to induce DVM in *Daphnia* (*Lass and Spaak, 2003*), and we have shown that the chemical characteristics of the kairomone are identical among different cyprinid species (*von Elert and Loose, 1996*) which matches the finding that α-CPS is the dominant bile salt in the Cypriniformes (*Hagey et al., 2010*).

We further demonstrate that the induction of DVM by TCDCA, a representative for taurine-conjugated $C_{24}$ bile acids, required substantially higher concentrations ($\geq 20$ μM, which equals $\geq 10$ mg dissolved organic carbon/L) than induction by α-CPS. The observation that free bile acids occur in fish feces suggests a low degree of bacterial deconjugation of conjugated bile acids by gut bacteria (*Philipp, 2011*). However, such bacterial deconjugation of TCDCA would not release DVM-inducing activity, as the resulting free $C_{24}$ bile acid CDCA was even less active than TCDCA. Neither this finding nor the fact that α-CPS, a fish metabolite that has not been microbially modified, constitutes the major kairomone in fish incubation water support the earlier suggestion that bacteria are involved in the production of the kairomone (*Ringelberg and Van Gool, 1998*).

As well non-cyprinid fish species like perch, stickleback (Perciformes) and pike (Esociformes) induce DVM in *Daphnia* (*von Elert and Pohnert, 2000*), and the chemical characteristics of kairomones from pike and stickleback match those of cyprinids (*von Elert and Pohnert, 2000*), which suggests that the kairomones of non-cyprinid and cyprinid species are similar and perhaps identical. However, a general pattern that emerges from a comprehensive analysis of bile salt patterns in fish indicates that α-CPS is confined to Cypriniformes and absent in the fish orders Perciformes, Esociformes and Salmoniformes. Yet, we here demonstrate the presence of α-CPS in bile of rainbow trout (*Oncorhynchus mykiss*), a representative of Salmoniformes. Still, this finding is in accordance with the general bile salt pattern in fish, as this pattern considers only bile salt types present at 5% or greater of the bile salt pool (*Hagey et al., 2010*). Furthermore, the authors argue, that the production of bile alcohols such as α-cyprinol (a $C_{27}$ bile alcohol) and their conjugation to sulfate is regarded as an evolutionarily ancient bile salt profile (*Hagey et al., 2010*). All evolutionarily more recent bile salt patterns of fish seem to have evolved from the putatively ancestral pattern of $C_{27}$ bile alcohol production by subsequent enzymatic modifications of them (*Hagey et al., 2010*; *Hofmann et al., 2010*). In line with this bile of many ray-finned fish species still contains traces of the biosynthetic precursor $C_{27}$ bile alcohol though their bile is dominated by conjugated $C_{24}$ bile acids (*Hagey et al., 2010*). We deduce that even in fish orders with a more recent bile acid pattern, α-CPS may be present in traces (i.e. <5%) as we have demonstrated for salmon, although this remains to be demonstrated for example for stickleback and Perciformes.

Our findings thus suggest that *Daphnia* evolution has selected for a high sensitivity to the evolutionarily oldest bile product of fish, that is α-CPS, and that this has proven to be an evolutionarily stable strategy as evolutionarily more recent fish bile patterns still contain traces of α-CPS. Thus, by sensing a single bile compound only, *Daphnia* are able to perceive the presence of fish across very different orders (*Hagey et al., 2010*). Future research will have to prove that 5α-CPS or other, hithertounidentified compounds, serve as the DVM-inducing kairomone as well in non-cyprinid fish orders.

The identification of α-CPS as the DVM-inducing kairomone in *Daphnia* allows for the assessment of its in situ concentration in space and time and thus to understand trait-variation in *Daphnia* and within-lake variations of DVM-amplitudes. The finding that a bile salt from fish serves as an infochemical might stimulate testing for a role of these compounds in the marine environment, where similar patterns of DVM of zooplankton are common. Still, although cessation of DVM in marine copepods in the absence of fish has been shown (*Bollens and Frost, 1991*), its induction in the presence of fish could not be demonstrated (*Bollens and Frost, 1991*). Similarly, a potential role of bile salts from fish in other anti-fish defenses in freshwater invertebrates remains to be tested, for example in changes in *Daphnia* life-history (*Stibor, 1992*) or morphology (*Tollrian, 1994*), in the reduced pigmentation of copepods in the presence of fish (*Hansson, 2000*) or in mediating effects of fish on the oviposition site of female *Chaoborus* (*Berendonk, 1999*). Knowing the kairomone now allows for

identification of the α-CPS binding receptor in *Daphnia* in order to understand the rapid evolution of DVM in response to the absence/presence of fish (*Cousyn et al., 2001*).

## Materials and methods

### Bioassay

We utilized a plankton organ (*Loose et al., 1993*) for the assessment of the vertical daytime distribution of daphnids with a setup according to *Brzeziński and von Elert (2015)*. The standardized experimental design has been shown to be a reliable indicator for the assessment of the kairomone, as the daytime depth determined on day 3 of the bioassay is a function of kairomone concentrations (*Loose et al., 1993*). Briefly, plexiglas tubes (length: 1 m long, volume: 200 mL, stoppered at the bottom) were placed vertically in a thermally stratified water bath and illuminated from the top (photoperiod 16L:8D). The thermal gradient in the organ (*Figure 1—figure supplement 1*) simulated a thermally stratified lake in summer. The tubes were either filled with control tap water to which the respective volumes of solvents had been added ('Control') or with control tap water to which different extracts had been added. The green alga *Acutodesmus obliquus* was used as food at saturating concentrations. Each treatment was replicated four- or fivefold with replicates and treatments being randomized.

Cohorts of second, third or fourth clutch neonates of *Daphnia magna* clone B, which has been isolated from a lake where it coexisted with fish (*Lampert, 1991*), were raised in tap water at saturating concentrations of *A. obliquus* until day 5 and then used to initiate the experiments. Five randomly chosen individuals of *D. magna* were transferred to each tube of the plankton organ and were fed with *A. obliquus* to a concentration of 2 mg C $L^{-1}$ every day. The vertical position of the animals in the tubes was determined by visual inspection with 5 cm accuracy. The mean daytime residential depth of the animals on the third or fourth day of incubation were taken as the measure of activity (for further details see *Loose et al., 1993*).

### Statistics

Data on daytime residential depth of *Daphnia* were analyzed for treatment effects by one-way ANOVA and Tukey's HSD posthoc test, if data met the assumption of homoscedasticity (Levene's test). Differences between treatments were considered to be significant at $p < 0.05$. Data from independent experiments were pooled if daytime depth for control and for EFI did not differ. Statistical analyses were computed by using the software R version 3.3.3 (R Core Team 2017) and the packages agricolae version 1.2–8 (2017) and car (2011).

### Kairomone extraction

The enrichment of fish incubation water was carried out for incubation water of *Rutilus rutilus*. Six pre-starved (24 hr) individuals (body size 10–20 cm) were incubated in 16 L of tap water at 18°C for 24 hr. The water was removed and filtered <0.65 µm. The kairomone was enriched from the water by $C_{18}$ solid-phase extraction (SPE) (Mega Bond Elut, $C_{18}$-bonded silica, mass: 75 g, Agilent Technologies) as according to *von Elert and Loose (1996)*. Briefly, 4 liters of the incubation water were adjusted to 1% methanol and passed through the cartridge. After a washing step with 1% methanol, the cartridge was eluted with 200 mL of methanol, and this eluate was evaporated to dryness and re-dissolved in methanol to yield 80 µL of extracted fish incubation water (EFI). It has been shown earlier that controls for $C_{18}$-SPE do not affect the daytime position of *D. magna* (*von Elert and Loose, 1996*).

### Extraction of fish bile and purification of 5α-cyprinol sulfate

Gall bladder of *Oncorhynchus mykiss* (rainbow trout) was homogenized in methanol with a pestle and then centrifuged. After quantification of the α-CPS content in the particle-free supernatant by LC-MS, aliquots were used in the bioassay that resulted in 0.4 and 4 nM α-CPS. These concentrations were close to the α-CPS concentration in active EFI (1 nM). Carp bile was extracted from the gall bladder, and bile salts were extracted with modifications as according to *Denton et al. (1974)*. 0.5 mL of bile were extracted by adding 10 mL of a hot 95:5 (v/v) mixture of ethanol and methanol and subsequent vortexing. After cooling, the supernatant was separated by centrifugation and

diluted with ultrapure water to an alcoholic concentration of 10% and then subjected to a $C_{18}$ solid-phase cartridge (Bond Elut, $C_{18}$, Agilent Technologies). The activated solid phase was preconditioned with 10% methanol, and the processed bile sample (corresponding to 0.5 mL of bile) was passed through the cartridge. After washing (10% methanol), the cartridge was eluted with methanol. The eluate was evaporated to dryness and dissolved in 1.5 mL methanol (extracted carp bile) corresponding to a 1:3 dilution of the raw bile. α-CPS was purified from this extract by HPLC fractionation as according to *Hahn et al. (2018)*. While 20% of the solvent leaving the LC column was transferred to the ESI-MS, the remaining 80% were at the same time collected by a fraction collector. In this way a fraction which contained α-CPS only was obtained. To eliminate co-eluting taurine-conjugated bile acids, the extracted carp bile was digested by choloylglycine hydrolase prior to chromatography. The collected fraction containing purified α-CPS was subjected to solid-phase extraction to remove the mobile HPLC phase. The identity of the isolated α-CPS was confirmed by NMR (*Hahn et al., 2018*), and its purity was demonstrated by liquid chromatography coupled to ESI-MS. LC-MS was performed as presented below in 'Identification of bile salts' using a reversed-phase (12.5 × 2 mm Nucleosil C18, 100–3, Macherey-Nagel, Düren, Germany).

## Fractionation of extracted fish incubation water

EFI was separated into six fractions by HPLC using the mobile phases water (A) and a mixture of acetonitrile (VWR, Radnor, Pennsylvania) and methanol (VWR, Radnor, Pennsylvania) (13:6, v/v) (B). Both solvents were made up to 0.015% (v/v) formic acid (Honeywell, Morristown, USA) and 10 mM ammonium acetate (Merck, Darmstadt, Germany). The following solvent gradient was applied to a 250 mm x 4 mm column (Nucleosil 100–5 $C_{18}$, Macherey-Nagel, Düren, Germany) at a flow rate of 1 mL/min with the column oven set to 30°C: 0 min: 35% B, 1.5 min: 35% B, 2.1 min: 40% B, 9 min: 45% B, 9.6 min: 55% B, 16.5 min: 65% B, 19.5 min: 100% B, 25.5 min: 100% B, 25.8 min: 35% B, 33 min: 35% B. 80% of the HPLC-derived eluate were collected by a fraction collector, while 20% were analyzed by MS. Chromatography and high-resolution mass spectrometry were conducted using an ultra-high pressure liquid chromatography (UHPLC) system (Thermo Fisher) with an Accela 1250 psi pump coupled with an ExactiveOrbitrap mass spectrometer (MS). Analytes were ionized by ESI in the negative mode. Ions within a mass range from *m/z* 120 to 1500, were detected at a frequency of one scan per second. The flow rate of the sheath and the aux gas nitrogen were set to 35 and 5 arbitrary units of the device. Spray voltage: 4.3 kV, capillary voltage −60 V, tube lens voltage −200 V, skimmer voltage −20 V, capillary temperature 325°C. The ExactiveOrbitrap mass spectrometer was calibrated once a day with the Pierce LTQ ESI negative calibration mixture. The instruments were controlled using the Xcalibur software (Thermo Scientific).

## Quantification of bile salts

Bile salts were quantified by HPLC coupled to MS as described above (Fractionation of EFI). Taurochenodeoxycholic acid (TCDCA) was quantified by the standard addition method (*Figure 4—figure supplement 3A*), while the concentrations of α-CPS were derived from a calibration curve using the purchasable cholesteryl sulfate (*Figure 4—figure supplement 3B*), whose molecular structure suggests a similar detectability by ESI-MS. The detected MS signals of all analytes were normalized to the detected signal of glycocholic acid, which was added to the samples as standards prior to the LC-MS analysis to account for variability of the mass spectrometer's sensitivity. All analytes were measured in the matrix of EFI in triplicates to compensate for its effects on MS measurements. The reference compounds sodium taurochenodeoxycholate (CAS:6009-98-9), sodium cholesteryl sulfate (CAS:2864-50-8), chenodeoxycholic acid (CAS: 474-25-9) and glycocholic acid hydrate (CAS:1192657-83-2) were of >95% purity and obtained from Sigma-Aldrich. The evaluation of the chromatograms was performed using the Xcalibur software (Thermo Scientific).

## Identification of bile salts

Bile salts were identified by LC-MS taking into account their exact *m/z* ratios and retention times. For the identification of CPS, additional MS/MS analyses were performed. Chromatography was performed on an UltiMate 3000 UHPLC system (Thermo Fisher). Analytes were separated on a reversed-phase column (BEH $C_8$, 100mmx2.1mm, Acquity), with the column oven adjusted to 60°C. The mobile phase consisted of two solvents: (A) ultrapure water and (B) a mixture of acetonitrile

and 2-propanol (both VWR, Radnor, Pennsylvania) (7/3, v/v). Both solvents contained ammonium acetate (Merck, Darmstadt, Germany) at a concentration of 10 mM added in form of a 1 M stock solution (1%, v/v) and acetic acid (VWR, Radnor, Pennsylvania) at a concentration of 1% (v/v). For chromatography, the following gradient was applied: 0 min: 40% B, 1 min: 40% B, 8 min: 75% B, 9 min: 100% B, 12.1 min: 40% B, 14 min: 40% B, 16 min: 40% B. Mass analysis was conducted on a Q Exactive Hybrid Quadrupole-Orbitrap (Thermo Fisher) equipped with an ESI source. Negative ionization was achieved by application of 2.5 kV. Sheath, auxiliary and sweep gas flow rates were adjusted to 50, 13 and 3 arbitrary units of the device. The capillary temperature was set to 263°C and the auxiliary gas heater temperature to 425°C. For the MS/MS measurements, negatively charged ions of the exact $m/z$ of CPS (531.2997 $m/z$) were selectively fragmented applying collision energy of 50 eV. The evaluation of the chromatograms was performed using the Xcalibur software (Thermo Scientific).

### Database research

For the identification of the most likely candidates of the chemical identity of the kairomone, we used the software MassWorks (Cerno Bioscience) to extrapolate from the measured exact masses and isotope patterns in the active HPLC fraction of EFI to sum formulas of the unknown molecules. A database search for molecules with these sum formulas was performed using the online platform ChemSpider.

## Acknowledgements

We are very thankful to L Hagey for providing α-CPS as a reference compound used in preliminary experiments and to P Fink for valuable comments on an earlier version of this manuscript. The authors acknowledge the significant contributions by the Biocenter-MS facility, University of Cologne.

## Additional information

### Funding

No external funding was received for this work.

### Author contributions

Meike Anika Hahn, Data curation, Software, Investigation, Visualization, Methodology, Writing—original draft, Writing—review and editing; Christoph Effertz, Investigation, Methodology; Laurent Bigler, Resources, Methodology; Eric von Elert, Conceptualization, Resources, Data curation, Supervision, Investigation, Methodology, Project administration, Writing—review and editing

### Author ORCIDs

Meike Anika Hahn  https://orcid.org/0000-0001-5090-0849
Eric von Elert  https://orcid.org/0000-0001-7758-716X

### Decision letter and Author response

Decision letter https://doi.org/10.7554/eLife.44791.025
Author response https://doi.org/10.7554/eLife.44791.026

## Additional files

### Supplementary files

• Transparent reporting form
DOI: https://doi.org/10.7554/eLife.44791.021

## Data availability

All data generated or analyzed during this study are included in the manuscript and supporting files. Source data files have been provided for Figure 1, Figure 1—figure supplement 2, Figure 4, Figure 3—figure supplement 2, and Figure 4—figure supplement 6.

The following dataset was generated:

| Author(s) | Year | Dataset title | Dataset URL | Database and Identifier |
|---|---|---|---|---|
| Hahn M, Effertz C, Bigler L, von Elert E | 2019 | Data from: A bile salt from fish induces diel vertical migration in zooplankton | http://dx.doi.org/10.5061/dryad.5d69g86 | Dryad Digital Repository, 10.5061/dryad.5d69g86 |

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
