## [Decision Letter]

Thank you for submitting your article "A bile salt from fish induces diel vertical migration in zooplankton" for consideration by *eLife*. Your article has been reviewed by three peer reviewers, including Georg Pohnert as the Reviewing Editor and Reviewer #1, and the evaluation has been overseen by Ian Baldwin as the Senior Editor.

The reviewers have discussed the reviews with one another and the Reviewing Editor has drafted this decision to help you prepare a revised submission.

Summary:

The concept of a compound that has to be released by fish due to their metabolic activity and that signals their presence to few millimeter sized *Daphnia* who make avoidance reactions is intriguing. The manuscript describes therefore a breakthrough in aquatic kairomone research and clearly demonstrates the activity of a fish bile salt to induce diel vertical migration in Daphnia. It answers a long-standing question and opens up new perspectives in kairomone research. The knowledge about the active principle will further open up new perspectives in environmental and ecological research.

Essential revisions:

All reviewers agreed that the manuscript describes an important progress in the field and a solid piece of chemical ecology. However, several concerns about the presentation of the work were raised.

As can be seen in the detailed comments below all reviewers recommend a substantial focus of the discussion on core-results of the paper. Further, two reviewers recommend that the quantification of the kairomone should be better documented.

The title could be more precise; the compound that was elucidated could be mentioned – especially since a specific activity was observed.

The manuscript is already in itself convincing, but the discussion about anthropogenic infodisruption (Abstract and Discussion) seems to be farfetched. The specificity of the response and the discussion about amounts of potential info disruptors required to triggers a response makes it a quite speculative discussion. The core result is very strong; there is no need to extend it into speculation. I would suggest deleting the part of the Discussion paragraphs five and seven. Also, reviewer 2 states clearly "The infodisruption part is a bit overstated. If TCDCA from anthropogenic sources is really released with sewage in such high concentrations, it might induce DVM in the vicinity of the inflow. However usually it will be diluted and probably most lakes where sewage is flowing in contain fish anyhow. Thus, the proposed ‘far-reaching implications for ecosystem functioning and conservation management’ are not obvious."

Further in many places the authors discuss DVM in general, leading to the impression that 5α-CPS is, or might be, responsible for DVM worldwide. This is most probably not true. There is no indication that DVM in marine environments is chemically induced at all. Bollens and Frost tested chemical cues from fish in enclosures and did not find an induction of DVM. While it is widely accepted that DVM is an anti-predator strategy, behavioral plasticity has not been shown in the marine environment. Furthermore, there are no cyprinids in most marine environments. To emphasize my point, this does not mean that DVM in the marine environment is not induced by 5α-CPS – it simply needs to be shown.

The authors suggest that 5α-CPS is the only relevant fish kairomone and their results indeed indicate that also rainbow trout produce this kairomone. However, Hagey et al., 2010, presented a number of orders which do not produce 5α-CPS, including Perciformes and Esociformes, but also Salmoniformes. Perciformes and Esociformes have been used to induce depth selection behavior, e.g., Ringelberg and van Gool, 1998, and the authors induced with *Esox lucius* in an earlier study. Thus, there seems to be some deviating results which require some discussion. It might simply indicate that some clarification is necessary, or it might call for future studies. The authors discuss effects on the marine carbon pump and suggest that 5α-CPS is the responsible cue. This is overstating.

Quantification is certainly the weakest point of the study: First normalization to cholesteryl sulfate and CPS would require a response factor of both analytes or better a calibration curve. The sentence “As 20 µL […]” suggests that the concentration of DPS was determined by bioassays not be analytical methods? Given the experimental I thought quantification with normalization to cholesterol sulfate was done. Also, reviewer 2 states: “It is necessary to present the concentrations clearer. E.g., the size of the fish should be mention as the release of the kairomone might be correlated to fish size. How large where the fish for gall bladder extraction? How much bile has been extracted in Rainbow trout?”

The reviewers also discussed the validity of the term bioassay guided fractionation: The approach used is known as "bioassay-guided fractionation", by which compounds purified from fish exudates were tested for escape-inducing behavior, resulting in the eventual identification of a single molecule responsible for the observed effect on prey. The manuscript is largely focused on the process of identifying this molecule via a combination of chemical and behavioral experiments. The molecule, 5α-cyprinol sulfate, is a known product of the digestive systems of fish – a logical candidate for a cue that prey might use to sense their predators and respond defensively. Given the fact that the authors just published a paper on this compound and its NMR data makes us wonder if we are not rather dealing with a targeted study confirming the activity of a logical candidate. This would twist the whole presentation but maybe leave a better feeling after reading. The current manuscript represents one such study. The fact that the molecule the authors identified was previously known (albeit present and active at low nanomolar concentrations), makes the chemistry of the study a little less exciting, but the biological implications are still important.

Reviewer 3 is quite critical in this point: The paper is solid and clear, but it lacks the kind of general appeal that expected for publication in *eLife*. In particular, the Introduction is quite basic, written in a way that is unlikely to draw in strangers to chemical ecology. The Discussion is similarly restricted to similar systems (fish and plankton). The description of the impacts of predation, chemical cues, and predator escape strategies would need broadening and enriching to be appropriate for this kind of general journal. The chemistry-oriented results are not particularly accessible to a general biology readership. As currently presented, the work is more appropriate for a more specialized journal like J. Chem. Ecol. or one committed to publishing works about freshwater ecology such as Limnology and Oceanography. The section of the Discussion related to movement of zooplankton (and thus effects on their own prey, phytoplankton) and resulting impacts on nutrient cycling would be of particular interest to a limnological and oceanographic audience. After a discussion among the reviewers the study was considered to be important and appealing to warrant further consideration of a revised version of the manuscript but the concerns of reviewer 3 should be kept in mind during revision and re-drafting of the Introduction and Discussion.

Introduction section / Discussion paragraph two: The statement “virtually nothing is known about the chemical nature […]" might have been true 10 years ago, however there is substantial progress in the field and while only few compounds have been fully investigated (these should be cited) several studies address the compound classes, functional modes of action etc. involved. This part of the Introduction should be expanded. This is a common notion among the reviewers "The authors suggest that virtually no aquatic kairomone has been identified, ignoring the recent publication of the *Chaoborus* kairomone – another relevant kairomone for *Daphnia* with some similarities (Weiss et al., 2018)."

Results second paragraph: The calculation is incorrect. The value 532.29993 would suggest that H has a mass of 1. It is however 1.007 will lead to a calculated mass of 532.30693.

Results second paragraph: Why is a data base search required to determine the sum formula from a high res mass? The software of the instrument can simply calculate this.

In the same paragraph: The step from the high res mass to the structure is not clear. Is the entire process only based on this one MS?

Figure 3: Are there explanations for the peaks at 2.04 and 2.38?

Figure 4: The structures are the major result of the paper – they could be plotted bigger. The stereocenter on the C/D ring carbon of CPS should be indicated. Is there any information about the stereochemistry in the side chain?

Results paragraph five: The negative ionization ESI MS in Figure 4—figure supplement 1 can by no means be seen as a proof of purity. In fact, negative ionization only picks up a fraction of compounds in a mixture and in addition, the MS is not clearly indicating a pure compound. Show at least chromatograms in positive and negative ionization mode if no NMR data are available.

Discussion paragraph six and onwards: The Discussion loses focus. Results from freshwater systems (this paper) are mixed with marine/oceanographic facts and become very speculative. The paper should focus more. Constructing e.g. a story about ocean acidification from the results is not justified.

Also: Salmon louse responses to CPS have to be proven, deduction from an uncharacterized receptor sequence to a DVM response are not warranted. Basically, a speculation is built on a speculation.

*Reviewer #2*

Some studies are ignored that proved that UV-radiation is also inducing DVM in Daphnia, showing that it is additionally a strategy leading to avoidance of harmful irradiation (see. Rhode et al., 2001 or Leech and Williamson (2001, L and O).

Ringelberg and van Gool, 1998, suggest that bacteria are responsible at a certain step of the kairomone production as addition of an antibiotic prevented depth selection behavior. This might be addressed in the Discussion.

Specific comments:

I suggest changing the title to indicate that currently 5α-CPS has been shown to induce DVM in *Daphnia magna* and most likely in all daphnids. There is no indication that other genera respond to this kairomone.

"A bile salt from fish induces diel vertical migration in *Daphnia*"

---

## [Author Response]

Essential revisions:All reviewers agreed that the manuscript describes an important progress in the field and a solid piece of chemical ecology. However, several concerns about the presentation of the work were raised.As can be seen in the detailed comments below all reviewers recommend a substantial focus of the discussion on core-results of the paper. Further, two reviewers recommend that the quantification of the kairomone should be better documented.The title could be more precise; the compound that was elucidated could be mentioned – especially since a specific activity was observed.

We have accordingly extended/changed the title into: “5α-cyprinol sulfate, a bile salt from fish induces diel vertical migration in Daphnia” (see also below, response to specific comments from #2).

The manuscript is already in itself convincing, but the discussion about anthropogenic infodisruption (Abstract and Discussion) seems to be farfetched. The specificity of the response and the discussion about amounts of potential info disruptors required to triggers a response makes it a quite speculative discussion. The core result is very strong; there is no need to extend it into speculation. I would suggest deleting the part of the Discussion paragraphs five and seven. Also, reviewer 2 states clearly "The infodisruption part is a bit overstated. If TCDCA from anthropogenic sources is really released with sewage in such high concentrations, it might induce DVM in the vicinity of the inflow. However usually it will be diluted and probably most lakes where sewage is flowing in contain fish anyhow. Thus, the proposed ‘far-reaching implications for ecosystem functioning and conservation management’ are not obvious."

As suggested, we now have removed this aspect from the Discussion and also from the Abstract.

Further in many places the authors discuss DVM in general, leading to the impression that 5α-CPS is, or might be, responsible for DVM worldwide. This is most probably not true. There is no indication that DVM in marine environments is chemically induced at all. Bollens and Frost tested chemical cues from fish in enclosures and did not find an induction of DVM. While it is widely accepted that DVM is an anti-predator strategy, behavioral plasticity has not been shown in the marine environment. Furthermore, there are no cyprinids in most marine environments. To emphasize my point, this does not mean that DVM in the marine environment is not induced by 5α-CPS – it simply needs to be shown.

We fully agree, except for the very minor point that in Bollens and Frost, 1991, behavioural plasticity of the copepods was shown (the cessation of DVM after removal of predators), but, of course, no induction by fish-cues. We are no more referring to DVM on a global scale, but confine the introduction to DVM in Daphnia. In the Introduction we are now stating: “One classical example of behavioural predator avoidance is diel vertical migration (DVM) in the freshwater microcrustacean Daphnia, which play a key role in lakes and ponds, as they are major consumers of planktonic primary producers and important prey for higher trophic levels. DVM is a widespread predator avoidance behavior […]”.

The authors suggest that 5α-CPS is the only relevant fish kairomone and their results indeed indicate that also rainbow trout produce this kairomone. However, Hagey et al., 2010, presented a number of orders which do not produce 5α-CPS, including Perciformes and Esociformes, but also Salmoniformes. Perciformes and Esociformes have been used to induce depth selection behavior, e.g., Ringelberg and van Gool, 1998, and the authors induced with Esox lucius in an earlier study. Thus, there seems to be some deviating results which require some discussion. It might simply indicate that some clarification is necessary, or it might call for future studies. The authors discuss effects on the marine carbon pump and suggest that 5α-CPS is the responsible cue. This is overstating.

We have now excluded any reference to the marine carbon pump. We are now addressing this issue of potential deviations in the Discussion in detail. In short, we argue (i) that the general phylogenetic pattern did not consider bile acids with <5% abundance, (ii) that we have shown that salmon has traces of a-CPS though it should not have according to the general phylogenetic pattern, which (iii) renders it possible that a-CPS serves as DVM-inducing kairomone as well in non-cyprinid fish oders, which (iv) remains to be confirmed by future research (Discussion paragraph eight).

Quantification is certainly the weakest point of the study: First normalization to cholesteryl sulfate and CPS would require a response factor of both analytes or better a calibration curve. The sentence “As 20 µL […]” suggests that the concentration of DPS was determined by bioassays not be analytical methods? Given the experimental I thought quantification with normalization to cholesterol sulfate was done.

α-CPS was enriched from fish incubation water by solid-phase extraction. The extract (EFI) corresponded (concerning the volume) to 50,000 times concentrated fish incubation water (density: 3 fish in 8 L water). Thus dissolving 20µL of this 50,000 times concentrated fish incubation water extract in 1 L corresponded to 1 L of the original non-processed incubation water (ignoring losses during extraction) (subsection “Statistics”). The concentration of α-CPS in the extract (EFI) and (thereby also in the artificially produced fish incubation water: EFI + aged and filtered tap water) have not been determined by bioassay. Instead, α-CPS in EFI was determined analytically by LC-MS using a calibration curve for cholesteryl sulfate in the background/matrix of EFI to exclude effects of ion suppression that might be caused by the complex matrix of EFI. We now point out that we only quantified α-CPS as equivalents of cholesteryl sulfate:

“[…] the concentrations of α-CPS were derived from a calibration curve using the purchasable cholesteryl sulfate (Figure 4—figure supplement 6), whose molecular structure suggests a similar detectability by ESI-MS.”

We further discuss the accuracy of this approach:

“Concerning the discussion of the estimated threshold concentration of α-CPS for DVM induction one methodological detail has to be considered. α-CPS present in EFI was quantified as equivalents of cholesteryl sulfate, since there has not been enough material of α-CPS to conduct an absolute gravimetric quantification. The quantification of α-CPS by means of a calibration curve of cholesteryl sulfate can however be considered quite accurate since both molecules probably result in comparable detector responses of the ESI-MS for the following reasons: i) both α-CPS and cholesteryl sulfate carry a sulfate group and, without prior ionisation, are already negatively charged when reaching the ESI source. Different ionizabilities of both analytes can thus be excluded as source of inaccuracy. ii) The m/z ratios of α-CPS and cholesteryl sulfate are so close to each other (m/z=465.3039 vs. m/z=531.2992), that tuning of the mass spectrometer’s ion optics will not differently affect the detectability of either compound. iii) Just like the m/z ratios, the molecular masses of the molecules are comparable as well, so that differences with respect to spray formation in the ESI source are probably of minor importance. In conclusion it is reasonable to assume that quantification of α-CPS as equivalents of cholesteryl sulfate results in minor inaccuracies only.”

Also, reviewer 2 states: It is necessary to present the concentrations clearer. E.g., the size of the fish should be mention as the release of the kairomone might be correlated to fish size. How large where the fish for gall bladder extraction? How much bile has been extracted in Rainbow trout?

We acknowledge the point that size of fish might be related to release of the kairomone and have now included this information in Material and Methods “body size 10– 20 cm”. We do have information about the size of the fish used for gall bladder extraction (2 mL of bile from a trout of 1 kg weight). Please note, that not all bile extracted from a single fish but instead only aliquots of this were used. We therefore felt that neither information on body mass nor on extracted bile volume would make sense and have therefore not included this information. However, if the reviewers require it, we are of course willing to include the above give information on weight and total bile extracted from an individual fish.

The reviewers also discussed the validity of the term bioassay guided fractionation: The approach used is known as "bioassay-guided fractionation", by which compounds purified from fish exudates were tested for escape-inducing behavior, resulting in the eventual identification of a single molecule responsible for the observed effect on prey. The manuscript is largely focused on the process of identifying this molecule via a combination of chemical and behavioral experiments. The molecule, 5α -cyprinol sulfate, is a known product of the digestive systems of fish – a logical candidate for a cue that prey might use to sense their predators and respond defensively. Given the fact that the authors just published a paper on this compound and its NMR data makes us wonder if we are not rather dealing with a targeted study confirming the activity of a logical candidate. This would twist the whole presentation but maybe leave a better feeling after reading. The current manuscript represents one such study. The fact that the molecule the authors identified was previously known (albeit present and active at low nanomolar concentrations), makes the chemistry of the study a little less exciting, but the biological implications are still important.

We acknowledge this point of concern. Below we give a detailed chronology of our approach hoping to convince the reviewers that we have performed a full bioassay-guided approach; it has by no means been a targeted-approach! Briefly, earlier chromatography of the extracted fish incubation water (EFI;using water-MeOH only) resulted in not well-focused biological activity spreading across fractions “Pre”, “8” and “9” (Author response image 1), so that going for less steep gradients with even less focused activity was no option.

**Author response image 1. respfig1:** Chromatogram and biological activity of extracted fish incubation water (EFI). (**a**) Chromatogram of EFI after separation on a 250 mm x 4 mm reversed phase column (Nucleosil 100-5-C_18_, Macherey-Nagel, Düren, Germany) using the mobile phases A H_2_O and B MeOH with the portion of B increasing over time. The total ion current of ESI-MS measurements in the negative ionization mode are depicted. (**b**) Behavioral response of *Daphnia magna* to extract of fish incubation water (EFI) and fractions thereof. Fraction numbering corresponds to the time of the chromatogram [min], while fraction “Pre” contains the collected eluate from 0-8 min and fraction “Post” that from 13-25 min. The control contains the same volume of pure organic solvent as tested from EFI and its fractions. Mean *Daphnia* daytime residence depth (+/- SD, n=4). Different capital letters indicate significant differences among treatments one-way ANOVA followed by Tukey post-hoc). Statistical results are summarized in Supplementary file 1.

We therefore investigated mass-scans spanning these three fractions “Pre”, “8” and “9”, which yielded one common prominent ion with m/z = 531.2941, (see Author response image 2, Author response image 4, Author response image 6). Comparisons of these three mass scans to corresponding mass scans from a previous blank run confirm that two masses can be neglected since they are constitutively detected in the blank run and that of EFI: 1) m/z=68.9942 (highlighted in red) and 2) m/z=112.9836 (highlighted in green). Excluding these contaminants after blank subtraction resulted in m/z=531.2941 to be the most prominent detected ion in fractions “Pre” (Author response image 2) and “8” (Author response image 4) and the one with the third highest signal detected in fraction “9” (Author response image 6).

**Author response image 2. respfig2:** ESI-MS data of the HPLC fractions leading to daytime depths of *Daphnia magna* which do neither statistically differ from the negative nor from the postive control and the corresponding spectra of a blank run. ESI-MS spectrum extracted from the time window corresponding to fraction: Author response image 2“Pre”, Author response image 4“8”, Author response image 6“9” in Author response image 1 and Author response image 3, Author response image 5, Author response image 7 the corresponding mass spectra from a previous blank run measured in the negative ionization mode. *MS/MS of m/z=531.3 lead to the appearance of a product ion of m/z=96.9598 which indicated the presence of a sulfate in the precursor ion (*Author response image 8).

**Author response image 3. respfig3:** 

**Author response image 4. respfig4:** 

**Author response image 5. respfig5:** 

**Author response image 6. respfig6:** 

**Author response image 7. respfig7:** 

**Author response image 8. respfig8:** MS/MS of m/z =531.3 in fraction “8”. Negative ionization MS/MS of m/z 531 [M– H]^–:^96.95979 (HSO_4_^–^_calc_=m/z 96.96010, ∆ = –3.2ppm), which indicated the presence of a sulfate group in m/z 531.30048 [M– H]^–^.

Both pieces of information, the accurate mass and the presence of sulfate, yielded 5α-CPS as a candidate. Since 5α-CPS is not commercially available, we asked Dr. Hagey who kindly provided reference material, which actually was material that he had obtained from his predecessor, Dr. Haslewood, and which was without any information concerning its purity or even its structural elucidation. We received too little material for NMR of this putative reference compound. Although this material from Dr. Hagey showed identical HR-mass and retention time with our compound (see Figure 3B,C in the manuscript) we still wanted to be sure about its chemical identity. Therefore, we isolated new material form carp bile and confirmed its structure by NMR. At that time, we decided to publish this structural elucidation as a separate paper, since it was a confirmation of an already known structure. Then, when being sure, that m/z=531.2941 represents 5α-CPS, we screened for further bile salts in fractions “Pre”, “8” and “9” and discovered that additional bile salts were present.

From there on we optimized our chromatography for the analysis of bile compounds, then using a different solvent system. This new mobile phase allowed for much more focused biological activity and much better resolution of bile compounds than with water-MeOH as mobile phases. It is these results that we now present in the manuscript. We would like to emphasize that at all stages of the research (as well in the part depicted in the paper) all fractions were always subjected to bioassays and the major activity was always considered/investigated further. We strongly regard this as a bioassay-guided approach, and, in line with this, we claim to have identified the major DVM-inducing compound. At no stage this approach has been a targeted approach.

Reviewer 3 is quite critical in this point: The paper is solid and clear, but it lacks the kind of general appeal that expected for publication in eLife. In particular, the Introduction is quite basic, written in a way that is unlikely to draw in strangers to chemical ecology. The Discussion is similarly restricted to similar systems (fish and plankton). The description of the impacts of predation, chemical cues, and predator escape strategies would need broadening and enriching to be appropriate for this kind of general journal. The chemistry-oriented results are not particularly accessible to a general biology readership. As currently presented, the work is more appropriate for a more specialized journal like J. Chem. Ecol. or one committed to publishing works about freshwater ecology such as Limnology and Oceanography. The section of the Discussion related to movement of zooplankton (and thus effects on their own prey, phytoplankton) and resulting impacts on nutrient cycling would be of particular interest to a limnological and oceanographic audience. After a discussion among the reviewers the study was considered to be important and appealing to warrant further consideration of a revised version of the manuscript but the concerns of reviewer 3 should be kept in mind during revision and re-drafting of the Introduction and Discussion.

We appreciate this remark and have thoroughly incorporated the issues raised. However, we admit, that it is not entirely clear what is meant by ‘lacks the kind of general appeal that is expected for publication in *eLife*’. Lakes and ponds are the best available freshwater source on the Earth's surface. Lakes are valued as water sources and for fishing, water transport, recreation, and tourism and, due to their outstanding importance, issues of management of lakes and ponds are of pivotal importance. One tool for lake management is food chain manipulation, which aims at manipulating the interaction of *Daphnia* and fish. Our paper provides a fundamentally new understanding of the interaction of *Daphnia* and fish, which is key to the functioning and ecosystem services of lakes and ponds, and we therefore would like to emphasize that we regard our findings as highly appealing for more general journals as *eLife*. In response to concerns raised we have provided two new first paragraphs in the Introduction, in which we introduce the general importance of predation, the distinction of direct and indirect effects of predators and the importance of chemical signalling in predator-prey interactions. We hope that this will more appropriate to draw in general biologists into an issue of chemical communication. We have further condensed statements about the chemistry of known kairomones in aquatic chemical communication from formerly three different paragraphs to a single paragraph starting with “The DVM-inducing kairomone that we report here is to […]”. We have further removed all aspects of nutrient recycling from the Discussion in order to be more focused on the main contribution of this manuscript. And we have now included an introductory definition for kairomone: “Such chemical cues are termed kairomones, if they mediate a transfer of information among species that impart a benefit to the receiving organism while not being beneficial for the producer”.

Introduction section / Discussion paragraph two: The statement “virtually nothing is known about the chemical nature […]" might have been true 10 years ago, however there is substantial progress in the field and while only few compounds have been fully investigated (these should be cited) several studies address the compound classes, functional modes of action etc. involved. This part of the Introduction should be expanded. This is a common notion among the reviewers "The authors suggest that virtually no aquatic kairomone has been identified, ignoring the recent publication of the Chaoborus kairomone – another relevant kairomone for Daphnia with some similarities (Weiss et al., 2018)."

We acknowledge this point. First, we would like to add that Weiss et al., 2018 had already been mentioned in the letter to the editor, but unfortunately not been included in the manuscript. We apologize for this. In response to this issue we have now included the few chemically identified kairomones in aquatic plankton and briefly discuss their chemical features. We are now stating in the Introduction “[…] recently progress has been made by identification of infochemicals involved in predator-prey chemical communication (Selander et al., 2015 Weiss et al., 2018, Poulin et al., 2018)”

In the Discussion we are now referring to the few identified compounds in more detail (the papers of Selander et al., 2015, Weiss et al., 2018, Poulin et al., 2018 and Uchida et al., 2008) and discuss chemical similarities starting with “The other few cases in zooplankton, for which chemical cues have been identified […]”.

Results second paragraph: The calculation is incorrect. The value 532.29993 would suggest that H has a mass of 1. It is however 1.007 will lead to a calculated mass of 532.30693.

Yes, we agree. We changed the m/z from 532.29992 to 532.3070.

Results second paragraph: Why is a data base search required to determine the sum formula from a high res mass? The software of the instrument can simply calculate this.

Yes, indeed we calculated it using the tool sCLIPS included in the software MassWorks (cerno BIOSCIENCE). We mention the procedure in the Results: “The tool sCLIPS included in the software MassWorks (cerno BIOSCIENCE) was used to predict the corresponding sum formula of the detected molecule based on the detected peak of its monoisotopic mass as well as its isotope pattern. The predicted sum formula for [M-H]- was C27H48O8S. A data base search using the platform PubChem yielded as best hit cyprinol sulfate (CPS), a known bile compound of fish as a candidate compound for the DVM-inducing activity. This finding was well in accordance with the detected sulfate group.”

In the same paragraph: The step from the high res mass to the structure is not clear. Is the entire process only based on this one MS?

We agree. The structure was elucidated not only using the high res mass. Now, we present the procedure step by step in the Results (see the point above).

Figure 3: Are there explanations for the peaks at 2.04 and 2.38?

The identity of the ion detected at 2.04 min is unknown. The peak at 2.38 min is most probably derived from the isomer β-cyprinol sulfate, since *Oncorrhynchos mykiss* is known to synthesize β-bile alcohols (Hagey et al., 2010). There was no reference compound available to confirm its identity by retention times. We mention these peaks as follows:

“While the ion detected at 2.04 min is of unknown identity (Figure 3 D), we speculate that the second peak in the EIC of m/z=531.29986 at 2.38 min (Figure 3 E) is the isomoer β-cyprinol sulfate.”

Figure 4: The structures are the major result of the paper – they could be plotted bigger. The stereocenter on the C/D ring carbon of CPS should be indicated. Is there any information about the stereochemistry in the side chain?

Yes, we agree. We now also depict the missing protons, indicating the stereocenter on the C/D ring Carbon of CPS and enlarged the plots of all structures. (Figure 4) Unfortunately, there is no information available on the stereochemistry of the side chain.

Results paragraph five: The negative ionization ESI MS in Figure 4—figure supplement 1 can by no means be seen as a proof of purity. In fact, negative ionization only picks up a fraction of compounds in a mixture and in addition, the MS is not clearly indicating a pure compound. Show at least chromatograms in positive and negative ionization mode if no NMR data are available.

Yes, we agree. Following your request, we now replaced Figure 4—figure supplement 1 by chromatograms of the purified α-CPS in positive and negative ionization mode, as well as chromatograms of blank runs for comparison (Figure 4—figure supplement 1). Additionally, we inserted mass spectra spanning the chromatogram’s time window which is exhibiting peaks in the α-CPS but not in the blank runs to seek for impurities (Figure 4—figure supplement 2).

Discussion paragraph six and onwards: The discussion loses focus. Results from freshwater systems (this paper) are mixed with marine / oceanographic facts and become very speculative. The paper should focus more. Constructing e.g. a story about ocean acidification from the results is not justified.Also: Salmon louse responses to CPS have to be proven, deduction from an uncharacterized receptor sequence to a DVM response are not warranted. Basically, a speculation is built on a speculation.

We agree and have removed both aspects from the Discussion.

Reviewer #2Some studies are ignored that proved that UV-radiation is also inducing DVM in Daphnia, showing that it is additionally a strategy leading to avoidance of harmful irradiation (see. Rhode et al., 2001 or Leech and Williamson (2001, L and O).Ringelberg and van Gool, 1998, suggest that bacteria are responsible at a certain step of the kairomone production as addition of an antibiotic prevented depth selection behavior. This might be addressed in the Discussion.

We are now addressing this aspect in the Discussion by stating:

“Neither this finding nor the fact that α-CPS, a fish metabolite that has not been microbially modified, constitutes the major kairomone in fish incubation water support the suggestion that bacteria are involved in the production of the kairomone (Ringelberg and van Gool, 1998).”

Specific comments:I suggest changing the title to indicate that currently 5α-CPS has been shown to induce DVM in Daphnia magna and most likely in all daphnids. There is no indication that other genera respond to this kairomone."A bile salt from fish induces diel vertical migration in Daphnia"

Accordingly, we have changed the title into: “5α-cyprinol sulfate, a bile salt from fish, induces diel vertical migration in Daphnia”.